

# Can the Pinus sylvestris var. mongolica sand-fixing forest develop sustainably in a semi-arid region ?

Yiben Cheng[1,2], Hongbin Zhan[3], Mingchang Shi[1]

[1]School of soil and water conservation, Beijing Forestry University, Beijing, China, 100083
[2]Forest Ecosystem Studies, National Observation and Research Station, Jixian, Shanxi, China, 042200
[3]Department of Geophysical, Texas A&M University in College Station, Texas, USA, 77840

*Correspondence to*: Yiben Cheng (chengyiben07@gmail.com)

**Abstract:** Desertification is a global environmental and societal concern at present, and China is one of the countries that face the most severe damage of desertification. China's so-called Three North shelterbelt Program (3NSP) has produced a vast area of lined forest in the semi-arid regions with the purpose of battling desertification. Such a wind-breaking and sand-fixing forest has successfully slowed down the incursion of desert. However, the vast artificial forestry consumes a large amount of water resources, which profoundly affect the fragile ecological environment in the semi-arid regions. In turn, a large amount of water loss also causes a great number of vegetation deaths or defects. To understand the water balance and sustainable development of artificial forest in semi-arid region, this study uses the 30-year-old lined Pinus sylvestris var. mongolica sand-fixing forest in the eastern part of Mu Us Sandy land in Northwestern China as an example. Specifically, this investigation studies the redistribution of water in soil under existing precipitation conditions, so as to evaluate whether the rain-feed forestry can develop sustainably or not. Rain gauge, newly designed lysimeter and soil moisture sensor are used to monitor precipitation, deep soil recharge (DSR) and soil water content, resulting in an accurate estimation of annual moisture distribution of the rain-feed Pinus sylvestris var. mongolica. The study shows that there are two obvious moisture recharge processes in an annual base for the Pinus sylvestris var. mongolica forest soil in Mu Us Sandy land: 1) the snow melted water infiltration-recharge process in the spring, and 2) the precipitation-recharge process in the summer. The recharge depth of the first process is 160 cm. The second process results in DSR (referring to recharge that can reach a depth more than 200 cm and may eventually replenish the groundwater reservoir). The DSR of 2016-2018 is 1.4 mm, 0.2mm, 1.2 mm, respectively. To reach the recharge depths of 20 cm, 40 cm, 80 cm, 120 cm, 160 cm, and 200 cm, the corresponding precipitation intensities have to be 2.6 mm/d, 3.2 mm/d, 3.4 mm/d, 8.2 mm/d, 8.2 mm/d, and 13.2 mm/d, respectively. The annual evaporation amount in the Mu Us Sandyland Pinus sylvestris var. mongolica forest is 426.96 mm in 2016, 324.6 mm in 2017, 416.253 mm in 2018. This study concludes that under the current precipitation conditions, very small but observable DSR happened, thus the groundwater system underneath the forest may be replenished, meaning that the artificial Pinus forestry can probably develop sustainably. This study confirms that developing limited amount forestry in semi-arid regions is likely in a sustainable fashion. The widely variable annual precipitation in semi-arid areas may affect this conclusion and should be investigated in the future.





**Keywords:** Pinus sylvestris var. Mongolia, Infiltration, Semi-arid region, Afforestation, Desertification control

## 1 Introduction

Desertification is a global threat that impacts heavily on the livelihoods of millions of people inside and outside the desert land (Yang et al., 2005). The true cost of desertification is frequently underestimated due to the unknown scale of these external and downstream impacts (Cao et al., 2011). Sand and dust storms happen when strong winds

impact the arid and semi-arid regions (Sun et al., 2015;Wang et al., 2010a). Sandstorms occur relatively close to the ground and can rise to kilometers high into the atmosphere and transport very long distances (Wu et al., 2013). Sandstorms impact human health, agriculture and transport (Wang et al., 2013). Agroforestry is a proven approach to battle desertification or even eradicate sandstorms in some areas, but agroforestry taxes heavily on water resources in arid and semi-arid regions, which may cause other unwanted environmental problems (Garc á Chevesich et al., 2017).

Soil moisture is a vital component in the ecological environment and is the source of life for plants on the earth surface (Sharma and Sharma, 2005). On the land surface of the Earth, not only the natural vegetation distribution is limited by water supply, but also the production of artificial vegetation relies on water supply more than any other factors (Gerten et al., 2004;Contreras et al., 2011). Therefore, the dynamic relations among soil, vegetation and soil moisture and the study on their regulation mechanism are of great concern in various disciplines like forestry,

agriculture, livestock, and environment (Young, 1989;Li et al., 2013b). For instance, the research in this field in China in the past decade has deeply influenced the economic development strategies of arid and semi-arid regions (Cao et al., 2011), and lessons learnt can be applicable for management of arid and semi-arid regions in other parts of the World as well.

Semi-arid regions are transitional zones between arid-regions and humid-regions, and are also active areas where

desertification takes place (Gong et al., 2004;Rhee et al., 2010). A semi-arid region generally refers to an area with a drought index (annual evaporation/annual precipitation) between 1.5 and 3.49. A region with less than 200 mm annual precipitation is often designated as an arid region (Su et al., 2007). A region with an annual precipitation between 200 mm and 500 mm is designated as a semi-arid region (Barton et al., 2008). The total arid and semi-arid area of the world is approximately 48 million km$^2$, which is a third of the entire continental area of the World, covering more

than 50 counties and regions (Gao et al., 2014). The vast arid, semi-arid, and semi-humid and drought-prone areas, are the birthplace of the world's ancient civilizations, and have an important place in the modern farming and livestock production industries (Ngigi et al., 2005). In the United States alone, 17 states in the West, which are mainly of arid and semi-arid climate, provide more than 80% of the farming and livestock products for the country (Powell et al., 1879). The former U.S.S.R. Central Asian arid and semi-arid regions provide 45% of the total commodity food for

the former U.S.S.R (Bouwman et al., 2005;Kraemer et al., 2015). The farming and livestock products from the Central and Western Australian arid and semi-arid regions play an important role in the world market as well (Fagg and Stewart, 1994).



China is one of the major arid countries in the world, with a total arid area of 2.8 million km$^2$ (Wang et al., 2002), and a total semi-arid and semi-humid and drought-prone areas of 2.13 million km$^2$ (Wang et al., 2004). These areas are mainly located to the north of the Kunlun Mountains-Qinling-Huai River line, starting from the northwestern border between China and Russia, all the way to the western foot of Daxinganling. The areas cover 965 counties in 16 provinces, municipalities, and autonomous regions (Su et al., 2007). This is more than half of the total land territory of China. In these areas, there are 55 million Acre arable land, which are 51% of the total arable land of China. 65% of these areas are arid land without irrigation conditions (Boserup, 2011). Especially in the vast areas along the Great Wall to the east of Baotou, Inner Mongolia, and in the Loess Plateau, 99.7% of the arable land are arid land. 83% of the arid and semi-arid areas in China are concentrated in the northwest (Ma et al., 2008). Therefore, drought is the typical climate characteristic and a natural concern in the northwest regions of China.

To control desertification in the northwest regions, China launched a so-called Three North Forest Program (3NSP) in 1978 (Wang et al., 2010b), a large-scale reforestation program in the Northeast, Northwest, and North China. At the end of the 20$^{th}$ century, the arid area in China expanded approximately 10,400 km$^2$ per year. Such a rapid expansion of arid area has been halted since the beginning of 21$^{th}$ century, partially thanks to the success of 3NSP. Nowadays, the trend of arid area expansion in China has been reversed and the arid area decreases 2424 km$^2$ annually. The sandy land (which is one of the most important types of arid area) in China used to expand 3436 km$^2$ annually at the end of the 20$^{th}$ century, and now it decreases 1980 km$^2$ annually (Wenhua, 2004;Jiao et al., 2012). Studies have shown that China has made great achievements in sand prevention and control. China and India have contributed one-third of the world's new vegetation coverage. 42% of China's new vegetation coverage area is forest, and 32% is agricultural land. The expansion of green areas in India is mainly due to the expansion of agricultural land (82%), and the contribution of forests is smaller than that in China (4.4%) (Bawa et al., 2010;Menon et al., 2002). However, even with such an extraordinary achievement, there are still grave concerns in managing windbreak and sand-fixation forests (Wang et al., 2010a;Cao et al., 2011;Wang et al., 2010b). For example, reforestation is not successful in some places with unclear reasons. Specifically, trees grow into dwarf trees, which are ineffective in battling desertification, or even die.

Pinus sylvestris var. mongolica is a geographical variant of European red pine in the Far East (Bao, 2015;Li et al., 2004). It is naturally distributed in large areas of China's humid and semi-humid areas, especially Daxinganling in the northeast. It inherits the original European red pine's adaptive ability to a variety of ecological environments, and has characteristics such as heliophile, drought-resistant, cold-resistant, and soil infertility-resistant. It is one of the most common species used in 3NSP (Zhu et al., 2006). Since the start of 3NSP in 1978, Pinus sylvestris var. mongolica has been introduced and planted on a large scale in the windy and sandy areas in the Northeast, Northwest, and North China (Runnström, 2000). Up to present, it has been applied in more than 300 counties across 13 provinces, municipalities and autonomous regions, with a total area over $3*10^5$ hm$^2$ (Hu et al., 2008).

Now the question becomes: Is the reforestation of Pinus sylvestris var. mongolica in battling desertification in semi-arid regions of China sustainable? We will try to answer this question from a hydrological point of view by inspecting the relationship of precipitation, soil moisture change, and deep soil recharge (DSR) (referring to recharge that can reach a depth more than 200 cm and may eventually replenish the groundwater reservoir). In particularly, if




a sizable DSR can occur, meaning that groundwater may be recharged from precipitation, the sustainable reforestation

in the region is possible. To accomplish this goal, this study uses a newly developed DSR lysimeter to monitor a 30-year old pine artificial forest in the Northwest China. The collected dataset is used to understand the soil moisture dynamic and the DSR of the Pinus sylvestris var. mongolica in sandy land. Specifically, we try to tackle the following issues: 1) Sources of soil water recharge in semi-arid areas, especially the source of deep soil layer moisture; 2) The precipitation density that causes infiltration and its maximal penetrating depth; 3) The rate of annual precipitation

infiltration; 4) The evaporation amount of the pine forestry land. The ultimate goal is to find out whether the rain-feed Pinus sylvestris var. mongolica sand-fixing forest can develop sustainably or not in the study site.

## 2 Material and Method

### 2.1 Overview of the study area

The area under study is located in Chagan Naoer, on the northeastern edge of Mu Us Sandy land (39°05′16.2″N,

109°36′04″E), as shown in Figure 1. Mu Us Sandy land mainly consists of semi-fixed and fixed sand dunes, adjacent with the Loess Plateau, located in a desert-loess transitional zone. It has northwestern wind in the winter with a typically dry winter climate and frequent sandstorm. It has southeastern monsoon in the summer. The summer climate is relatively humid, and it is easy to form local heavy precipitation. The multi-year average precipitation is 400 mm, mostly concentrated during the summer. The groundwater table depth varies between 2 m to17 m in Mu Us Sandyland,

and it is 8 m at the experimental area of this study (Runnström, 2003). The groundwater table is lower in the summer and higher in the spring, with a variation less than 1.5 m. Since the initiation of 3NSP at Mu Us Sandy land in 1989, Pinus sylvestris var. mongolica has been planted in lines in the experimental area, with a 10 m line spacing, an average plant height of 6 m, and an average crown diameter of 6.6 m. The seasonal frozen soil period in the experimental area is from January to April, and from November to December in an annual base (Li et al., 2013a).

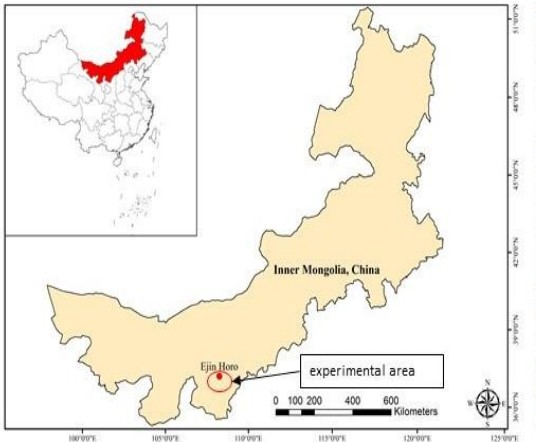
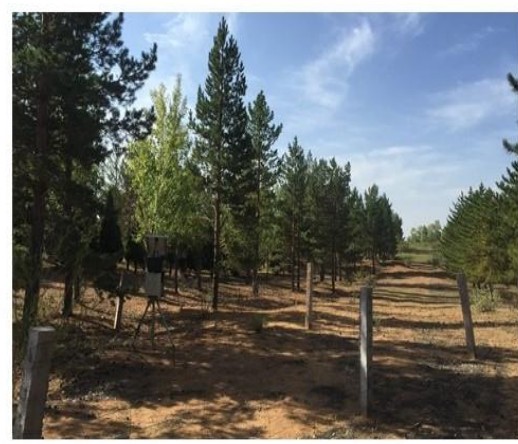


Figure 1. Geographic location of the experimental area and study site.



## 2.2 Experimental design

Root analysis of Pinus sylvestris var. mongolica shows that it has a shallow root distribution and underdeveloped main root, which belongs to a typical lateral root type. The 30-year-old Pinus sylvestris var. mongolica's root distribution can reach 6.5 m, with a concentrated area of 2.5 m. In this study site, the ground water level is too deep to supply for roots, so Pinus sylvestris var. mongolica root is mostly distributed over a vertical range of 0-0.5m and relies on precipitation for water supply. The original main root usually stops growing at 1.5-2 m depth (Zhu et al., 2006). Therefore, in the experimental design of this study, the lowest soil moisture sensor is placed at a depth of 200 cm, and the lysimeter placement is also at the 200 cm depth. The canopy of Pinus sylvestris var. mongolica is capable of intercepting precipitation, thus affects the measurements of precipitation and soil moisture directly underneath the canopy (Roth et al., 2007). Therefore, the measurements of this study are made in the middle between the forest lines, without the interference of canopy.

### 2.2.1 Soil moisture monitoring

Based on the root depth of Pinus sylvestris var. mongolica, the depth range of soil moisture sensor placement is determined. A soil section is cut out in the middle between two forest lines. The section consists of a layer of dead tree leaves, a leached layer, a depositional layer, and a native soil layer, which is of fine sand. Soil moisture sensors (MiniTRASE TDR, USA) are placed in soil layers at 20 cm, 40 cm, 80 cm, 120 cm, 160 cm, and 200 cm depths. The measurement interval is one hour.

### 2.2.2 DSR monitoring

To study the moisture distribution of Pinus sylvestris var. mongolica in Mu Us Sandy land, two sets of data need to be collected: precipitation from a rain gauge, and DSR measurement from a lysimeter. Surface runoff does not exist in the experimental area, thus is not a concern. Precipitation is monitored by rain gauge (Spectrum, USA, accuracy 0.2 mm), placed 1.5 m above ground surface. This study uses a new lysimeter to measure the DSR (Cheng et al., 2017). Such a new lysimeter has two parts: an upper water balance part and a lower measurement part. As shown in Figure 2, the water balance part uses a cylindrical impermeable side wall to enclose a soil column for measurement. The length of soil column is determined based by the capillary rise, which is approximately 60 cm, based on the soil particle size distribution at the site. The advantage of this design is that when soil at point B in Figure 2 reaches saturation, the capillary water reaches point A. Therefore, when infiltrated water enters the water balance part at depth A, additional infiltrated water after satisfying the saturation of soil between A and B will go into the measuring part. The measurement part has a measurement accuracy of 2 mm (Spectrum, USA). The lysimeter is placed at 200 cm depth to measure DSR, meaning that any precipitation-induced infiltration passing the 200 cm depth will not be subject to evaportranspirative process anymore. In another word, the infiltrated water that can pass the 200 cm depth of soil will keep going down and may become groundwater recharge.





Before taking the measurements, the new lysimeter needs to be placed one year in advance, going through naturally settlement for a year, and allowing the soil to reach its pre-installation condition.

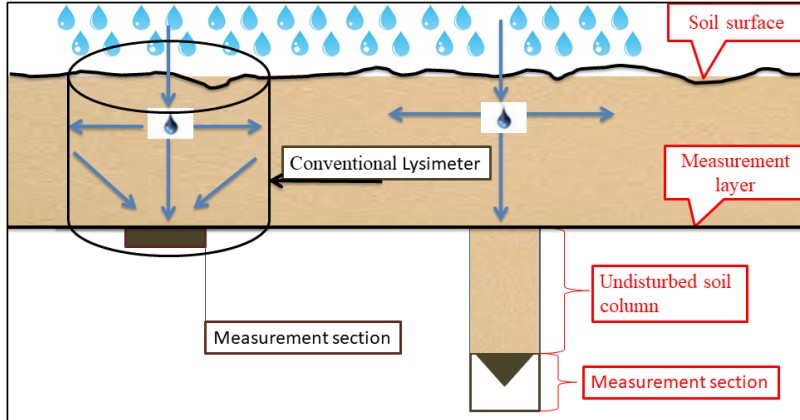

Figure 2. The schematic plot of a new lysimeter (on the right) with respect to the conventional lysimeter (on the left).

**3 Results and Discussion**

The soil moisture variation of Pinus sylvestris var. mongolica in 2016 is shown in Figure 3. It reveals that soil moisture has obvious seasonal variational trends. The soil from January to March is frozen. The near surface soil moisture recharge is from snowmelt. When the near surface frozen soil starts to thaw, soil at the 20 cm depth is recharged on February 9th, 16th and 26th in 2016. Soil at depths greater than 20 cm remains relatively stable. Frequent precipitation events usually occur from June to November, during which soil moisture changes considerably, and soil moistures at different depths exhibit periodic increase or decrease, regulated by the interplay of precipitation and evapotranspiration. After February 26th in 2016, soil gradually thaws completely. Figure 3 shows that snowmelt can recharge the soil moisture as deep as 140 cm. The soil moisture at 200 cm depth is recharged for the first time after a heavy precipitation event on July 8th in 2016.





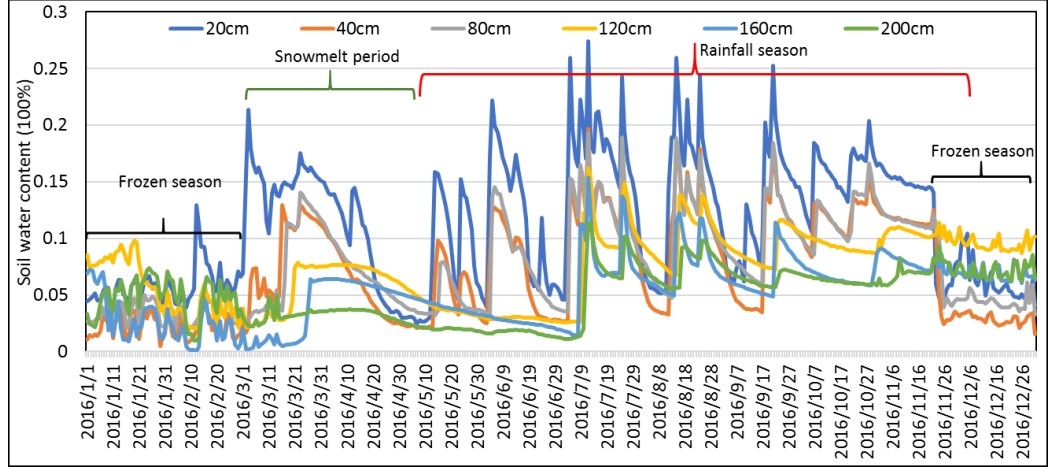

175          Figure 3. Annual precipitation and soil moisture of each layers in 2016.

In order to study the degree of soil moisture response to precipitation in individual layers, this research choose each layer's soil moisture at the beginning of each month as a representative, to observe whether the soil moisture in a specific layer is recharged. Figure 4 shows the soil moistures at depths of 20 cm, 40 cm, 80 cm, 120 cm, 160 cm, and 200 cm at the beginning of each month. From Figure 3, the soil of Pinus sylvestris var. mongolica exhibits four
180   distinctive layers: an evaporation layer at 0-40 cm depth, a lateral root activity layer at 40-160 cm depth, a dry soil layer at 160-200 cm depth, and a deep soil layer below 200 cm. For the 0-40 cm evaporation layer, the soil moisture increases only under the effect of precipitation or snowmelt. Its moisture content decreases rapidly under the interplay of evaporation and infiltration. For the 40-160 cm root activity layer, the soil moisture is recharged from infiltrated water passing through the upper layer, and it gradually decreases under the effects of infiltration and root moisture
185   absorption. For the 160-200 cm dry soil, the infiltrated water hardly reaches this layer, and the soil layer with a soil moisture under the withered point is formed. The deep soil below 200 cm depth is of native sand soil, and Figure 3 shows that the soil moisture content of this layer is recharged four times under heavy precipitation events in 2016.



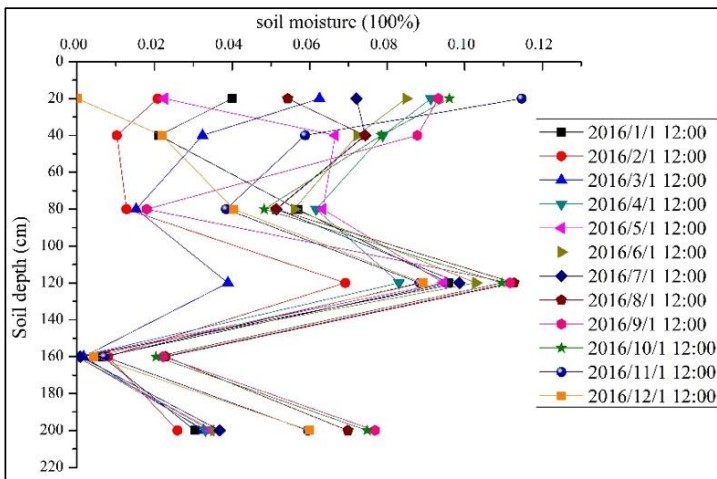

Figure 4. Month soil moisture changes of every soil layers.

The total precipitation in 2017 is 309 mm, which was a dry year. There are 32 times observed precipitation in the whole year. The maximum precipitation intensity mount is 22 mm/d, in July 22, 2017. The soil moisture fluctuation in 2017 is similar to that in 2016. There is a freezing period from January-March, the surface soil freezes, and the soil water content changes in each soil layers are relatively stable. March to April belongs to the freezing and thawing mixed period, frozen water in the soil layer gradually melts, especially the surface layer frozen water functions as a reservoir. The frozen water replenishes the soil moisture of each soil layer; from April to November, it belongs to the soil water active period of the rainy season. Under the combined action of precipitation recharge, soil evapotranspiration and vegetation consumption, the soil moisture fluctuates rapidly; then the soil begins to freeze again in December. As shown in Figure 5, the soil moisture fluctuations intensely in the soil layers at 20 cm, 40 cm, and 80 cm depths. The soil layer from 120 cm to 200 cm is relatively stable, and it is only replenished by freeze-thaw soil moisture during the freezing and thawing period (March to April). The 120 cm depth soil layer is replenished with water during the continuous summer precipitation process (on 14 October). The annual soil moisture infiltration is only 0.2 mm, which is 1.2 mm lower than that of 2016. The lack of precipitation in 2017 causes a sharp drop in deep soil moisture infiltration.





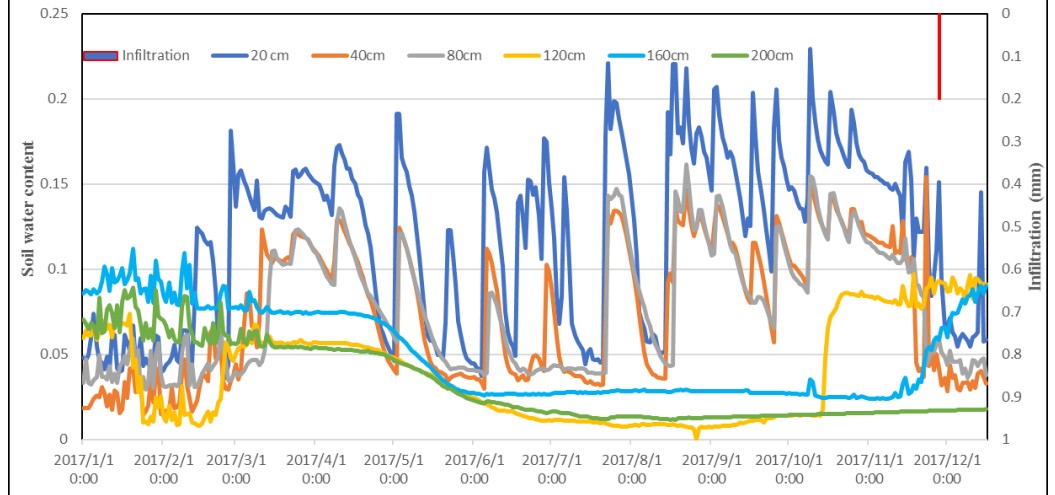

205                         Figure 5 Soil moisture and DSR dynamic change in 2017

        The total precipitation mount in 2018 is 472.2 mm, which is a wet year. There are 42 observed precipitation events
throughout the year with a maximum precipitation intensity of 20 mm/d on 21 July, 2017. The soil moisture fluctuation
in 2018 is similar to that in 2016 and 2017. There is a freezing period in January-March, in which the soil water content
changes in each layer are relatively stable. The March and April belong to the freezing and thawing period, the frozen
soil water gradually melts, and soil below the surface layer is replenished by the melted water. Consequently, the soil
moistures in various layers rise accordingly. From April to November, it belongs to the active season of the rainy
season. Under the combined action of precipitation recharge, soil evapotranspiration and vegetation consumption, the
soil moisture fluctuates. The shallow soil layer begins to freeze again in December.

        As shown in Figure 6, the layers of strong soil moisture fluctuations are 20 cm, 40 cm, 80 cm, 120 cm, and 160
cm. The 200 cm level of soil changes relatively small, and was only recharged on September 7th of 2018. The annual
soil moisture infiltration is only 1.2 mm in 2018, which is 1 mm higher than that in 2017 and 0.2 mm lower than that
in 2016.





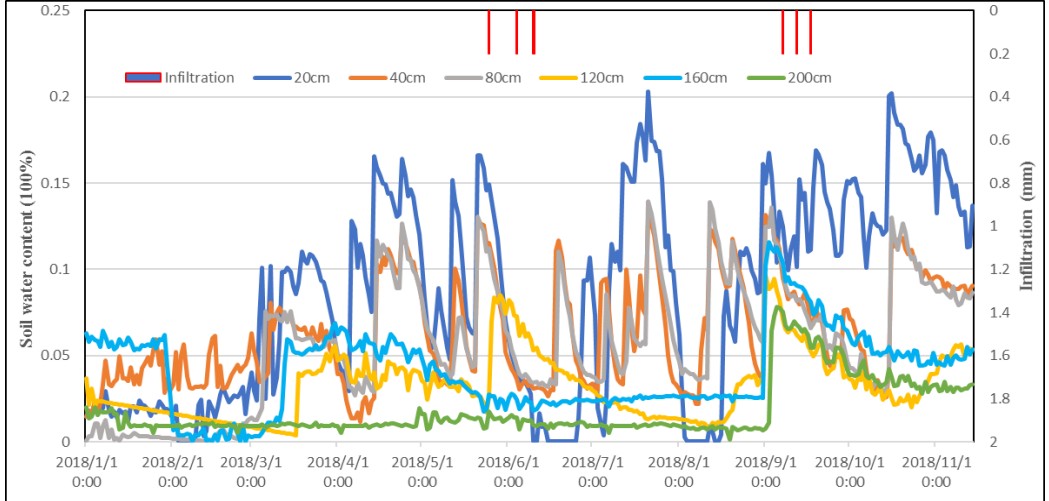

220                                     Figure 6. Soil moisture and DSR dynamic change in 2018.

**3.1 Soil moisture infiltration rate comparison during different seasons**

The soil moisture recharge sources in the experimental area are spring snowmelt and summer precipitation. According to Figure 3, we can clearly see that the soil water recharge in different seasons varies, especially at the end of winter season and in the summer rainy season. The amount of spring precipitation in this study site is small, and
snowmelt moisture is the main water source in this semi-arid area. The germination process of vegetation or seeds in semi-arid areas mainly depends on the water source of accumulated snowfall in winter. With the surface soil temperature increasing, the surface ice water gradually melts and infiltrates to deeper soil layer. As shown in Figure 7 (AB), this study chooses two typical processes for comparison: the snow melted soil moisture recharge process from February 26th to March 27th of 2016, and the precipitation recharge process from July 3rd to 12th of 2016. During the
process of snowmelt infiltration, the soil wetting front moves slowly downward in the vertical direction, as shown in Figure 5A, it takes 2 days and 7 hours for the wetting point reaching the 60 cm soil layer, but for summer precipitation infiltration process, it takes only 1 day for wetting front to reach 60 cm soil layer.

The February 26th to March 27th of 2016 snow melted soil moisture recharge process lasts for 29 days, and the soil moisture recharge depth reaches 160 cm. The soil moisture at 200 cm depth does not show any noticeable change,
suggesting that DSR has not been generated. The start time of moisture recharge is set at when soil moisture content starts to increase. The end time of moisture recharge is set at when soil moisture content reaches its maximum. These two moisture infiltration processes are shown in Figure 7. There are many factors affecting the rate of precipitation infiltration. A model that does not adequately consider the most relevant factors can certainly leads to erroneous simulation results. In the future, the gap between the experimental measurement results and the corresponding model
simulation results much be investigated and eventually filled.





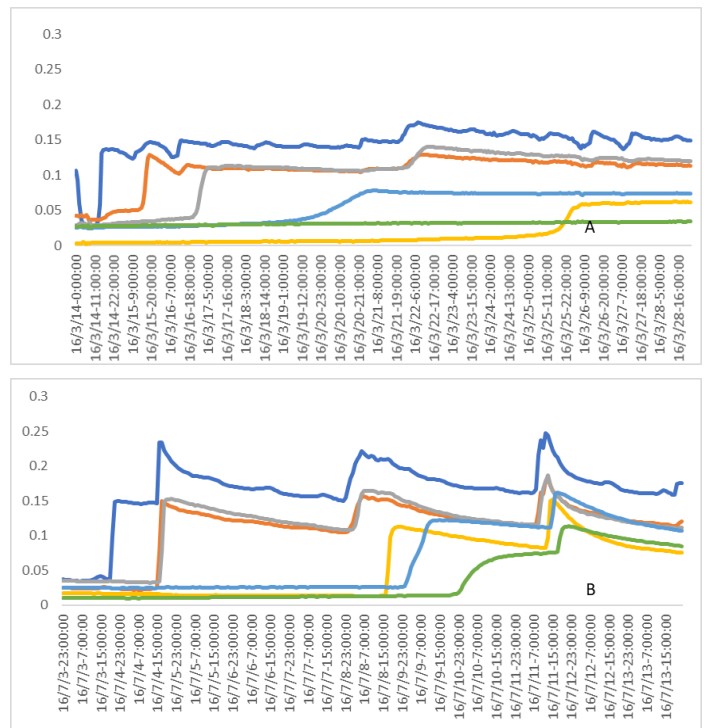

Figure 7. Two recharge process (snow melt process and precipitation process) in 2016.

### 3.2 Recharge intensity of different soil layer infiltration

In the laboratory, one may calculate the precipitation intensity infiltrating into a specific soil layer according to the soil characteristics with a proper mathematical model. In the natural environment, however, there are too many factors affecting the infiltration process, such as temperature and air humidity, wind speed, surface soil moisture, soil heterogeneity, etc. In this study, we computed the replenishment of each soil layer by analyzing the precipitation-induced wetting point to find out the minimum precipitation intensity for each individual layer.

Precipitation infiltration results in increasing soil moisture content. Each time when soil moisture infiltrates into a designated depth, it leaves a crest signal. Based on the comparison between the number of crest signals and the number of precipitations, the minimum precipitation amount can be determined by the crest signals at different soil depths. Statistics of precipitation data from 2016 to 2018 and fluctuations in soil moisture content in each layer are shown in Table 1, which shows that when precipitation infiltrates to soil layers at 20 cm, 40 cm, 80 cm, 120 cm, 160

cm, and 200 cm depths, the minimum precipitation intensities are 2.6 mm/d, 3.2 mm/d, 3.4 mm/d, 8.2 mm/d, 8.2 mm/d, and 13.2 mm/d, respectively. When heavy precipitation events (precipitation intensity higher than 100 mm/d) happen in the study site, precipitation can recharge the soil moisture at 200 cm depth.



Table 1 Precipitation produced moisture increase signal and corresponding minimum precipitation intensity (data from 2016-2018).

| Soil layer depths | Sum of soil moisture increase signals on each soil layer | Corresponding minimum precipitation intensity |
|---|---|---|
| 20cm | 74 | 2.6mm/d |
| 40cm | 46 | 3.2mm/d |
| 80cm | 32 | 3.4mm/d |
| 120cm | 16 | 8.2mm/d |
| 160cm | 16 | 10.2mm/d |
| 200cm | 10 | 13.2mm/d |

**3.3 Yearly moisture distribution of Pinus sylvestris var. mongolica**

Figure 3 shows that during the seasonal frozen-soil period, soil moisture is relatively stable. The monthly average values for soil moistures of different soil layers in January and December of 2016 are used as the start and end soil moisture values. Although the precipitation amount varies from 2016 to 2018, other environmental factors in this area are basically the same, and soil moistures are similar. To figure out the Pinus sylvestris var. mongolica water balance

from 2016 to 2018, one has:

$$P-DSR-ET=\delta W \qquad (1)$$

where P is precipitation, ET is evapotranspiration, and $\delta W$ is the whole 200 cm soil layer moisture change. Runoff is not included in above water balance equation because it does not occur during the experiment. As P, DSR, and $\delta W$ can be accurately measured, ET can be calculated by above equation.

270                    Table 2 Water distribution in 2016 of the rain-feed Pinus sylvestris var.

| Year | Precipitation | DSR | δW | ET |
|---|---|---|---|---|
| 2016 | 466.4mm | 1.4mm | 38.056mm | 426.96mm |
| 2017 | 309mm | 0.4mm | -16mm | 324.6mm |
| 2018 | 472.2mm | 1.2mm | 54.747mm | 416.253mm |

As shown in Table 2, precipitation in the experimental area in 2016 is 466.4 mm and the DSR is 1.4 mm. Thanks to the heavy precipitation in the summer and the higher than multi-year average precipitation (multi-year average precipitation is 400 mm), all soil layer moisture content ($\delta W$) increases 38.056 mm within the upper 200 mm depth. The groundwater table is 8 m depth, beyond the root range of Pinus sylvestris var. mongolica. Therefore, Pinus

sylvestris var. mongolica cannot absorb and utilize groundwater. Based on the real-time monitored precipitation, DSR



and soil moisture content change, the evapotranspiration of Pinus sylvestris var. mongolica can be calculated as 426.96 mm/year. 2017 is a dry year with a precipitation of 309 mm, DSR of 0.4 mm, soil water storage decreased by 16 mm, evapotranspiration mount is 324.6 mm. The precipitation mount in 2018 is 472.2 mm. The DSR is 1.2 mm, the soil water storage capacity is increased by 54.747 mm, and the evapotranspiration is 416.253 mm. Based on the redistribution data of precipitation in the shallow soil (200 cm depth) over the past three years, one can see that precipitation has a recharge effect on both shallow and deep soil layers. In the shallow soil layer, evapotranspiration in the dry year consumes stored water, but in the wet year precipitation water recharge the shallow soil layer. Deep soil layer infiltration has been recorded in the past three years with very small amount, indicating that under the existing vegetation cover and rain-fed conditions, the precipitation is barely able to support the shallow ecosystems, and only a small amount of precipitation can percolate into the deep soil layer.

Comparing the data of three years in Table 2, DSR of the experiment site is relatively small, indicating that the soil moisture resource in the Pinus sylvestris var. mongolica forest is limited under this rain-fed conditions. In the dry years, the soil evaporation was relative large, but the measured total evapotranspiration decreased, indicating that the transpiration of vegetation was inhibited. Under this condition, Pinus sylvestris var. cannot grow well. In semiarid regions, precipitation varies considerably every year, and the year of 2016 may not be representative of the long-term average behavior of DSR in this region as the precipitation of this year is higher than the average annual precipitation of 400 mm. Instead, it may be more representative of a wet year behavior, so as 2017 and 2018. To figure out the long-term behavior of DSR and SWC in the semiarid regions such as Mu Us Sandy land, one must carry out a multi-year (preferably a decade long) experiment.

## 4 Summary and Conclusions

As one of the four largest artificial shelter forest system, three North shelter forest system project which includes the Mu Us Sandy land investigated here, has a history of nearly 40 years of construction. On one hand, artificial shelter forest prevents the invasion of desert; on the other hand, construction of a vast artificial shelter forest may have detrimental effect on ecological environment in arid regions and may substantially change the evapotranspiration pattern in the region. Since precipitation is almost the only water sources for replenishing the groundwater system in the Mu Us Sandy land, change of evapotranspiration will greatly affect the groundwater recharge in the region, which directly determines if the reforestation can be sustainable. This study uses a 30-years-old mature Pinus sylvestris var. mongolica forest of a 10-meter line spacing as the target of the experiment. The study uses a new lysimeter to monitor DSR and to accurately calculate water balance from 2016 to 2018. The following conclusions can be drawn from this study:

1. Pinus sylvestris var. mongolica forest soil in Mu Us Sandy land has two significant moisture recharge processes in an annual base: spring snow melted moisture infiltration-recharge process and summer precipitation-recharge process. The recharge depth of spring snow melted moisture recharge process is 160 cm. The summer precipitation-recharge process results in DSR, recharging the soil moisture below 200 cm. The DSR of 2016-2018 is 1.4 mm, 0.2mm, 1.2 mm, respectively. Under the existing precipitation conditions, water supply in the





rain-fed pine forest can meet the consumption of vegetations but the remaining amount of rain-fed infiltration that can percolate into deep soil layer is small.

2. The experimental results show that the precipitation intensities are respectively 2.6 mm/d, 3.2 mm/d, 3.4 mm/d, 8.2 mm/d, 8.2 mm/d, and 13.2 mm/d when precipitation infiltrates into 20 cm, 40 cm, 80 cm, 120 cm, 160 cm, and 200 cm soil depths. Infiltration depth and precipitation intensity are not linearly related.

3. In semi-arid areas, the annual precipitation varies greatly, the dry and wet years alternate, DSR mount is relatively small, soil water mount is limited. The growth of Pinus is affected by annual precipitation. Compared to the start of the year, soil moisture content increases 38.056 mm, -16 mm, 54.747 mm.

4. The precipitations in 2016-2018 are 466.4 mm, 309 mm, 472.2 mm, and the associated DSR values are 1.4 mm, 0.2 mm, 1.2 mm, respectively. Under the current precipitation condition and reforestation design, the natural recharging moisture of Pinus sylvestris var. mongolica can meet the plant growth needs, and have additional moisture for DSR which may eventually recharge groundwater.

5. Calculation based on these dataset shows that the annual evaporation of Pinus sylvestris var. mongolica forest in Mu Us sandy land is 426.96 mm, 324.6 mm, 416.253 mm for year 2016-2018, respectively. Pinus automatically adjusts its evapotranspiration in response to different precipitation amount, and this may affect the development of Pinus sylvestris var. As extreme weather conditions happen more frequently worldwide (possibly due to the global warming effect), the arid region precipitation may change rapidly. Whether Pinus sylvestris var. mongolica can adapt to this trend is a question that still needs decades-long observational effort.

**Acknowledgements:**

This study was supported with research grants from the Fundamental Research Funds for the Central Universities (BLX201814) and the National Natural Science Foundation of China (41771306). I especially thanks Chinese Scholar Council support me to go to Texas A&M University as a visiting researcher.

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
