# Peer review of "Can the Pinus sylvestris var. mongolica sand-fixing forest develop sustainably in a semi-arid region ?"

_Hydrology and Earth System Sciences, 2019_

## Referee Comment (RC1) · Anonymous Referee #1 · 18 Apr 2019

Using a newly developed DSR lysimeter, the authors monitored the soil moisture dynamic and the DSR of the Pinus sylvestris in Mu Us Sandy land for 3 years (2016-2018). They found that there were two soil water recharge source with spring snow melting and summer precipitation. Also they concluded that the proportional precipitation intensities varied across recharge depth. This paper addresses a timely issue for san-fixing forest, given expected changes deep soil recharge. In general terms, the manuscript has a clear focus and in situ measurements (although the methods are simple) adequately tested the research questions herein. Sections of the manuscript, however, take away from the strengths of this manuscript, particularly the Results and Discussion. Thus, I proposed a few comments as the following. General Comments.

[Figure]

1. The Title: "Can the Pinus sylvestris var. mongolica sand-fixing forest develop sustainably in a semi-arid region". The topic was "too large", and, the paper seemed like to study the soil moisture dynamics and recharge source, not relevant the subject. I really think that the study is interesting, but the title might reflect better the performed research. 2. It was not appropriate to rely on three years (2016-2018) of soil moisture measurements to determine whether the sand-fixing tree species survives. First, the experiment time was too short, and the artificial trees with life cycles over decades. The adaptability of long-lived woody species cannot be based solely on water, temperature, light, soil texture, etc. 3. Logically describe the work you do in the Introduction. For example, the description of the semi-arid regions and the drought situation in China should be merged into other paragraphs without the need for separate sections; and these statements were not relevant to the subject of this study. 4. The Result and Discussion should be separated. I saw more results but no discussion. 5. For the Summary and Conclusions, it reads too much like the Abstract and simply restates the main results, instead of leaving the reader with a "take-home message" and "fruit for thought".

Special comments. 1. Need to mark the line number. 2. The main part of the Abstract focused on describing the research background (almost 1/3). I would expect some results and discussion or implications of the main findings. 3. L30. What do you mean the current precipitation conditions? Was it the annual precipitation (2016, 2017, and 2018) mentioned later? If so, what was the relationship between the evaporation and precipitation? 4. In the Keyword, I would recommend adding soil moisture and DSR, which were the two main monitoring indicators of this study. 5. L95. Replace with "over $3 \times 105$ hm2" 6. L155. "In another word" is not common and can be replaced with "In other word". 7. L170. The recharge depth of spring snow melting in Abstract was 160 cm (L25), why was here it 140 cm? Another problem was that I did not see the recharge depth in 2017 and 2018, and only in 2016. Can I think that the recharge depth in the Abstract was the 2016? 8. L 175. Why was there no change in precipitation in Figure 3? 9. L265. Move to the Methods. 10. In the Conclusions (No. 3), what was the

start of the year? 2016? why was there a negative value (-16 mm)?

---

## Author Comment (AC1) · 23 Apr 2019

We thank the reviewer for the constructive comments. The manuscript has been significantly improved by addressing the comments. The following are our point-to-point responses to their comments.

1. The Title: "Can the Pinus sylvestris var. mongolica sand-fixing forest develop sustainably in a semi-arid region". The topic was "too large", and, the paper seemed like to study the soil moisture dynamics and recharge source, not relevant the subject. I really think that the study is interesting, but the title might reflect better the performed research. Reply: Implemented. The title of the paper has been changed to "On the soil

moisture dynamics of sand-fixing Pinus sylvestris var. mongolica forest in a semi-arid region". We think this title is a better representation of the study.

2. It was not appropriate to rely on three years (2016-2018) of soil moisture measurements to determine whether the sand-fixing tree species survives. First, the experiment time was too short, and the artificial trees with life cycles over decades. The adaptability of long-lived woody species cannot be based solely on water, temperature, light, soil texture, etc. Reply: Implemented. The point raised in this comment is elaborated in the discussion section as follows. "One should be cautious that the three years (2016-2018) soil moisture measurements presented in this study may not always be reliable for performing long term (such as decades long) prediction of whether the studied species can develop sustainably over decades as some artificial trees may have life cycles over decades long. Therefore, continuous (preferably decades long) measurements are necessary in the future. Another notable point is that the adaptability of long-lived woody species may not be based solely on water, temperature, light, and soil texture. Despite of such limitations, we think this three-year investigation offers an important step for understanding the soil moisture dynamics of sand-fixing Pinus sylvestris var. mongolica forest in a semi-arid region. Furthermore, these three years happen to encompass rather dramatically different weather patterns in the region (wet versus dry years), thus offer additional insights on the function of the Pinus sylvestris var. mongolica forest under highly variable external forces."

3. Logically describe the work you do in the Introduction. For example, the description of the semi-arid regions and the drought situation in China should be merged into other paragraphs without the need for separate sections; and these statements were not relevant to the subject of this study. Reply: Implemented. The introduction has been reorganized. The description of the semi-arid regions and the drought situation in China has been substantially shortened as suggested.

4. The Result and Discussion should be separated. I saw more results but no discussion. Reply: Implemented. The Results and Discussion have been separated.

5. For the Summary and Conclusions, it reads too much like the Abstract and simply restates the main results, instead of leaving the reader with a "take-home message" and "fruit for thought". Reply: Implemented. The Summary and Conclusions have been revised to convene the "take-home message" and "fruit for thought"

Special comments. 1. Need to mark the line number. Reply: Only line numbers of every 5 lines are added according to the requirements of HESS.

2. The main part of the Abstract focused on describing the research background (almost 1/3). I would expect some results and discussion or implications of the main findings. Reply: The implemented. The research background has been shortened in the abstract.

3. L30. What do you mean the current precipitation conditions? Was it the annual precipitation (2016, 2017, and 2018) mentioned later? If so, what was the relationship between the evaporation and precipitation? Reply: Implemented. The current precipitation conditions refer to the three-year (2016-2018) precipitation conditions. We have revised the text for clarification.

4. In the Keyword, I would recommend adding soil moisture and DSR, which were the two main monitoring indicators of this study. Reply: Implemented.

5. L95. Replace with "over $3 \times 105$ hm2" Reply: Implemented.

6. L155. "In another word" is not common and can be replaced with "In other word". Reply: Implemented.

7. L170. The recharge depth of spring snow melting in Abstract was 160 cm (L25), why was here it 140 cm? Another problem was that I did not see the recharge depth in 2017 and 2018, and only in 2016. Can I think that the recharge depth in the Abstract was the 2016? Reply: Implemented. The issue has been clarified.

8. L 175. Why was there no change in precipitation in Figure 3? Reply: Implemented. The precipitation information is added in Figure 3.

9. L265. Move to the Methods. 10. In the Conclusions (No. 3), what was the start of the year? 2016? why was there a negative value (-16 mm)? Reply: Implemented. The starting year is 2016. The soil moisture storage change is the soil moisture storage at the end of the year minus the soil moisture storage at the beginning of last year. The negative value means that the soil moisture reserve has decreased by 16 mm during the year.

—————————————————

---

## Referee Comment (RC2) · Anonymous Referee #2 · 26 Apr 2019

Sustainable soil remediation is an important and urgent topic, as it is presently still unclear how effective some remediation strategies are. The authors address the use of Pinus sylvestris var. mongolica as a way to fixate sand in the Mu Us Sandy land in Northwestern China, specifically, if rain-fed forestry can sustainable develop in the region. Their study describes the use of a newly developed deep soil recharge lysimeter to monitor a 30-year old pine artificial forest. The current presentation of the methodology lacks sufficient detail (see specific comments below), and the results present only parts of the water balance on which the conclusions are then based. This results in a paper that is currently difficult to evaluate. Also, there is no discussion section present. Specific comments: Line 16: in this semi-arid region Line 18: "as an example" In the

introduction I read in line 90-99 that many of the reforestation efforts are unsuccessful and Pinus sylvestris var. mongolica (Psvm) is the most common specie used in 3NSP. There is no statement on whether this species is more resilient, or why the abstract mention this as being an example. Can these several statements be more connected to clarify the actual success of using Psvm? Line 29-30: Reported results are for period 2016-2018, and it is concluded deep soil recharge happened and thus Psvm can sustainably develop. I think this conclusion is not merited on a 3 year observation period. There is nothing reported on the state of the 30 year old Psvm forest, are these trees normally developed or not? Were they irrigated during that time? What is the minimum amount of water needed for these trees to survive? Also I have my doubts at the significance of the reported digits of the water balance. See later comments. Line 37: In the abstract the term desertification was used in the context of an arid environment. Here it is used in a broader sense. The World Atlas of Desertification (2018) has revised the definition due to confusion outside the context of (semi-)arid areas and now promotes the use of the term land degradation instead of desertification. Whether the authors use desertification or land degradation I suggest to refer to a formal definition in this particular general context. Line 101:"sustainable" This question is not very specific. Sustainable in term of what exactly? Line 128-129: Unclear if this refers to general observations from literature or from the inspected field site. Please explain. Line 135-137: "measurements were made without interference of canopy in the middle between forest lines" So, no root water uptake was measured, no intercepted precipitation or interecepted evaporation? The assumption being that the amount of deep recharge will be the amount available to the trees to take up water? But in that case you are missing the interception term, and the amount of water infiltration in the forest will be less than what is measurement in the DSR setup, which then invalidates your conclusions. Please explain more clearly so the reader can follow. Line 139-143: What was the soil composition/classification? Which soil moisture sensors were used? The miniTrase refers to the cable tester, not to the particular type of moisture sensor. Was a factory calibration used with the soil moisture sensors, and which one? Line 146: "surface runoff does not exist" Was this not observed, even in snowmelt conditions? What was the slope in the area? Please specify. Line 152-154: "point A and B" not indicated in Fig. 2 Fig 2: Does not include an arrow for evaporation? Is it not measured? Fig 3: The caption states the figure shows annual precipitation as well, but only the soil moisture readings are reported. As soil moisture changes in the lysimeter are a result of precipitation and soil evaporation (assuming no trees were growing in the lysimeter), so please include precipiatation over 2016-2018 in the results section. Also in the frozen season the soil moisture sensors readings drop. As freezing impacts the dielectric permittivity the sensor readings can be impacted. See for example: Hallikainen, M. T., Ulaby, F. T., Dobson, M. C., El-Rayes, M. A., & Wu, L. K. (1985). Microwave dielectric behavior of wet soil-part 1: Empirical models and experimental observations. IEEE Transactions on Geoscience and Remote Sensing, (1), 25-34. Nothing is mentioned in the text, regarding these readings. Line 186: Please explain how I can assess this from Fig 3? I see more than four increases in soil moisture at the 200 cm depth. Do the authors mean there are four time when water was collected in the measurement section of the DSR? Fig 4: Changes are reported in increments of <0.01%. What was the accuracy of the soil moisture sensors, and are the data in Fig 4 not impacted by this? Or is the scale perhaps not what I think I am seeing, fraction instead of percentage? Line 201: "annual soil moisture infiltration" Please explain how these reported numbers were obtained in the methodology, I suspect the infiltration is either weighed or measured otherwise, but it is not described at present. Table 2: If the accuracy of the precipitation measurement is 0.2 mm, the calculated ET cannot have more reported significant digits. Please adjust. Also if as stated the measurements were made in the middle between the forest line, what then caused the transpiration term in the ET? Or was the DSR placed underneath the canopy, while precipitation and soil moisture were measured outside of the canopy. For me as a reader it became a bit vague at this point. In this water balance the canopy interception is not considered as well, making the concluding statements doubtful. Line 213: And this would not depend on posterior soil moisture conditions as well? Line 318&324: Please adjust the reported significant

digits.

---

## Author Comment (AC2) · 2 May 2019

We thank the reviewer for the constructive and detailed comments. The manuscript has been significantly improved by addressing the comments. The following are our point-to-point responses to the comments.

Sustainable soil remediation is an important and urgent topic, as it is presently still unclear how effective some remediation strategies are. The authors address the use of Pinus sylvestris var. mongolica as a way to fixate sand in the Mu Us Sandy land in Northwestern China, specifically, if rain-fed forestry can sustainable develop in the region. Their study describes the use of a newly developed deep soil recharge lysimeter

to monitor a 30-year old pine artificial forest. The current presentation of the methodology lacks sufficient detail (see specific comments below), and the results present only parts of the water balance on which the conclusions are then based. This results in a paper that is currently difficult to evaluate. Also, there is no discussion section present.

Reply: Implemented. The focus of the study is to measure the deep soil moisture infiltration in the Pinus sylvestris var. mongolica forest land, using a newly designed lysimeter. The detailed description of the design and application of the instrument has been documented in a previous publication on HESS (Cheng et al., 2017), so it is only briefly described in this study. The Pinus sylvestris var. mongolica has been in existence in the study area for more than 30 years, so the purpose of this study is to find out whether there are sufficient water resource available in the region to support vegetation ecosystem, through the measurement of deep soil water recharge (or DSR). As suggested, the title of the paper has been changed to "On the soil moisture dynamics of sand-fixing Pinus sylvestris var. mongolica forest in a semi-arid region", which is a better representation of the body of this study. We have rewritten the discussion part as suggested.

Specific comments: Line 16: in this semi-arid region Line 18: "as an example" In the introduction I read in line 90-99 that many of the reforestation efforts are unsuccessful and Pinus sylvestris var. mongolica (Psvm) is the most common specie used in 3NSP. There is no statement on whether this species is more resilient, or why the abstract mention this as being an example. Can these several statements be more connected to clarify the actual success of using Psvm?

Reply: Implemented. Most of the afforestation practices mentioned in the introduction of the paper were terminated prematurely due to various reasons. In contrast, the Pinus sylvestris var. mongolica was found to be a suitable species in the practice of afforestation in the Three North Region of China. Therefore, the lessons leant from this afforestation practice are important for future adaptation of this species in other regions as well. We have revised the statements and provided further references to address

the issue of resilience of this species.

Line 29-30: Reported results are for period2016-2018, and it is concluded deep soil recharge happened and thus Psvm can sustainably develop. I think this conclusion is not merited on a 3-year observation period. There is nothing reported on the state of the 30-year-old Psvm forest, are these trees normally developed or not? Were they irrigated during that time? What is the minimum amount of water needed for these trees to survive? Also I have my doubts at the significance of the reported digits of the water balance. See later comments.

Reply: Implemented. The point raised here is similar to the comment No. 2 from the other reviewer. Please see our detailed response to that comment. "One should be cautious that the three years (2016-2018) soil moisture measurements presented in this study may not always be reliable for performing long term (such as decades long) prediction of whether the studied species can develop sustainably over decades as some artificial trees may have life cycles over decades long. Therefore, continuous (preferably decades long) measurements are necessary in the future. Another notable point is that the adaptability of long-lived woody species may not be based solely on water, temperature, light, and soil texture. Despite of such limitations, we think this three-year investigation offers an important step for understanding the soil moisture dynamics of sand-fixing Pinus sylvestris var. mongolica forest in a semi-arid region. Furthermore, these three years happen to encompass rather dramatically different weather patterns in the region (wet versus dry years), thus offer additional insights on the function of the Pinus sylvestris var. mongolica forest under highly variable external forces."

Line 37: In the abstract the term desertification was used in the context of an arid environment. Here it is used in a broader sense. The World Atlas of Desertification (2018) has revised the definition due to confusion outside the context of (semi-)arid areas and now promotes the use of the term land degradation instead of desertification. Whether the authors use desertification or land degradation I suggest to refer to a

formal definition in this particular general context. Line 101:"sustainable" This question is not veryspecific. Sustainable in term of what exactly?

Reply: Implemented. We have revised the language and the choice of words. For instance, we will use the term "land degradation", and we will re-define the term "sustainable" as meeting the growth needs of Pinus sylvestris var. mongolica and also having excess water to replenish deep soil layer as a sustainable standard.

Line 128-129: Unclear if this refers to general observations from literature or from the inspected field site. Please explain. Line 135-137: "measurements were made without interference of canopy in the middle between forest lines" So, no root water uptake was measured, no intercepted precipitation or interecepted evaporation? The assumption being that the amount of deep recharge will be the amount available to the trees to take up water? But in that case you are missing the interception term, and the amount of water infiltration in the forest will be less than what is measurement in the DSR setup, which then invalidates your conclusions. Please explain more clearly so the reader can follow.

Reply: Implemented. "Root analysis of Pinus sylvestris var. mongolica shows that it has a shallow root distribution and underdeveloped main root, which belongs to a typical lateral root type" This is the result of in situ observation after on-site excavation. Similar findings can also be found in the literature. In semi-arid areas, vegetation depends on precipitation, and roots are concentrated in shallow soil. Measuring the water absorption of roots, precipitation or interception of the canopy are complicated processes and often involves a great degree of uncertainty. This research, however, regards the canopy and topsoil as an integrated entity. And by measuring the amount of water entering the entity (precipitation) and the amount of water leaving the entity (deep soil recharge), one may conduct a water balance computation to calculate the amount of overall evapotranspiration.

Line 139-143: What was the soil composition/classification? Which soil moisture sensors were used? The miniTrase refers to the cable tester, not to the particular type of moisture sensor. Was a factory calibration used with the soil moisture sensors, and which one? Line 146: "sur-face runoff does not exist" Was this not observed, even in snowmelt conditions? What was the slope in the area? Please specify.

Reply: Implemented. The soil type in this area is sandy soil, the particle size distribution of 0-200 cm depth is as follows: extra coarse sand of 0.00%, coarse sand of 3.23%, middle sand of 50.53%, find sand of 36.06%, very fine sand of 7.19%, and silt sand of 2.99%. The EC-5 soil moisture probe was used and the correction equation is (li, 2012;Calibration of EC-5 Soil Moisture Sensors and Its Application in Arid Desertificated Area): . where xsand and ysand are Analog value and Standard value. The topographic variation of area is almost negligible and long-term observations show that there is no surface runoff. The amount of snow in the spring is not great enough to penetrate into the 140 cm soil layer.

Line 152-154: "point A and B" not indicated in Fig. 2 Fig 2: Does not include an arrow for evaporation? Is it not measured?

Reply: Implemented. The figure is revised as below

Fig 3: The caption states the figure shows annual precipitation as well, but only the soil moisture readings are reported. As soil moisture changes in the lysimeter are a result of precipitation and soil evaporation (assuming no trees were growing in the lysimeter), so please include precipiatation over 2016-2018 in the results section. Also in the frozen season the soil moisture sensors readings drop. As freezing impacts the dielectric permittivity the sensor readings can be impacted. See for example: Hallikainen, M. T., Ulaby, F. T., Dobson, M. C., El-Rayes, M. A., & Wu, L. K. (1985). Microwave dielectric behavior of wet soil-part 1: Empirical models and experimental observations. IEEE Transactions on Geoscience and Remote Sensing, (1), 25-34. Nothing is mentioned in the text, regarding these readings.

Reply: Implemented. We have revised the manuscript to include more detailed information on the soil moisture data and precipitation data of the past three years (2016-2018) to analyze the DSR. Under low temperature conditions in the winter, the accuracy of EC-5 may drop by 5% (according to the original manufacturer's instructions). To avoid the possibly unreliable data in the winter, we focus on analyzing the data from April to November (unfrozen ground period).

Line 186: Please explain how I can assess this from Fig 3? I see more than four increases in soil moisture at the 200 cm depth. Do the authors mean there are four time when water was collected in the measurement section of the DSR?

Reply: Implemented. According to Figure 3, the soil moisture content of the upper 200 cm soil layer fluctuates multiple times during the three-year experimental period. After November, the soil moisture content of the upper 200 cm soil layer fluctuations but DSR is not detected. This is probably due to the error of the EC-5 probe under frozen winter condition. Therefore, the active research period has been revised to from April to November each year. Between April to November, the DSR signals generated by the precipitations on August 11 and August 22 cannot be distinguished from each other, so we combine them together as one event. We will revisit this matter in this revised version to get a better description by inspecting the data more carefully. Fig 4: Changes are reported in increments of <0.01%. What was the accuracy of the soil moisture sensors, and are the data in Fig 4 not impacted by this? Or is the scale perhaps not what I think I am seeing, fraction instead of percentage?

Reply: Implemented. Figure 4 is revised to avoid the confusion. The data on the abscissa will be multiplied by 100%, with a range of 0-0.12, or 0-12% for better inspection.

Line 201: "annual soil moisture infiltration" Please explain how these reported numbers were obtained in the methodology, I suspect the infiltration is either weighed or measured otherwise, but it is not described at present. Table 2: If the accuracy of the precipitation measurement is 0.2 mm, the calculated ET cannot have more reported significant digits. Please adjust. Also if as stated the measurements were made in

the middle between the forest line, what then caused the transpiration term in the ET? Or was the DSR placed underneath the canopy, while precipitation and soil moisture were measured outside of the canopy. For me as a reader it became a bit vague at this point. In this water balance the canopy interception is not considered as well, making the concluding statements doubtful.

Reply: Implemented. "The annual soil moisture infiltration is only 0.2 mm, which is 1.2 mm lower than that of 2016." Should be revised as "The annual DSR measured at the 200cm soil depth is only 0.2 mm, which is 1.2 mm lower than that of 2016."

Line 213: And this would not depend on posterior soil moisture conditions as well? Line 318&324: Please adjust the reported significant digits.

Reply: Implemented. The line 210-213 describes the general state of soil moisture fluctuations in the experimental area after precipitation. The soil moisture rises with precipitation and decreases after vegetation consumption or infiltration. It is notable that the precipitation event in arid and semi-arid areas is not always strong enough to induce DSR. The line 318-314 refers to the change in soil moisture in 2016 and this part will be revised accordingly.
* * *
[Figure]

We thank the reviewer for the constructive and detailed comments. The manuscript has been significantly improved by addressing the comments. The following are our point-to-point responses to the comments.

*Sustainable soil remediation is an important and urgent topic, as it is presently still unclear how effective some remediation strategies are. The authors address the use of Pinus sylvestris var. mongolica as a way to fixate sand in the Mu Us Sandy land in Northwestern China, specifically, if rain-fed forestry can sustainable develop in the region. Their study describes the use of a newly developed deep soil recharge lysimeter to monitor a 30-year old pine artificial forest. The current presentation of the methodology lacks sufficient detail (see specific comments below), and the results present only parts of the water balance on which the conclusions are then based. This results in a paper that is currently difficult to evaluate. Also, there is no discussion section present.*

Reply: Implemented. The focus of the study is to measure the deep soil moisture infiltration in the Pinus sylvestris var. mongolica forest land, using a newly designed lysimeter. The detailed description of the design and application of the instrument has been documented in a previous publication on HESS (Cheng et al., 2017), so it is only briefly described in this study. The Pinus sylvestris var. mongolica has been in existence in the study area for more than 30 years, so the purpose of this study is to find out whether there are sufficient water resource available in the region to support vegetation ecosystem, through the measurement of deep soil water recharge (or DSR). As suggested, the title of the paper has been changed to "*On the soil moisture dynamics of sand-fixing Pinus sylvestris var. mongolica forest in a semi-arid region*", which is a better representation of the body of this study. We have rewritten the discussion part as suggested.

*Specific comments: Line 16: in this semi-arid region Line 18: "as an example" In the introduction I read in line 90-99 that many of the reforestation efforts are unsuccessful and Pinus sylvestris var. mongolica (Psvm) is the most common specie used in 3NSP. There is no statement on whether this species is more resilient, or why the abstract mention this as being an example. Can these several statements be more connected to clarify the actual success of using Psvm?*

Reply: Implemented. Most of the afforestation practices mentioned in the introduction of the paper were terminated prematurely due to various reasons. In contrast, the Pinus sylvestris var. mongolica was found to be a suitable species in the practice of afforestation in the Three North Region of China. Therefore, the lessons leant from this afforestation practice are important for future adaptation of this species in other regions as well. We have revised the statements and provided further references to address the issue of resilience of this species.

*Line 29-30: Reported results are for period2016-2018, and it is concluded deep soil recharge happened and thus Psvm can sustainably develop. I think this conclusion is not merited on a 3-year observation period. There is nothing reported on the state of the 30-year-old Psvm forest, are these trees normally developed or not? Were they irrigated during that time? What is the minimum amount of water needed for these trees to survive? Also I have my doubts at the significance of the reported digits of the water balance. See later comments.*

Reply: Implemented. The point raised here is similar to the comment No. 2 from the other reviewer. Please see our detailed response to that comment. "One should be cautious that the three years (2016-2018) soil moisture measurements presented in this study may not always be reliable for performing long term (such as decades long) prediction of whether the studied species can develop sustainably over decades as some artificial trees may have life cycles over decades long. Therefore, continuous (preferably decades long) measurements are necessary in the future. Another notable point is that

**Fig. 1.**

---

## Author Comment (AC4) · 18 May 2019

We thank the reviewer for the constructive and detailed comments. The manuscript has been significantly improved by addressing the comments. The following are our point-to-point responses to the comments.

Sustainable soil remediation is an important and urgent topic, as it is presently still unclear how effective some remediation strategies are. The authors address the use of Pinus sylvestris var. mongolica as a way to fixate sand in the Mu Us Sandy land in Northwestern China, specifically, if rain-fed forestry can sustainable develop in the region. Their study describes the use of a newly developed deep soil recharge lysimeter

to monitor a 30-year old pine artificial forest. The current presentation of the methodology lacks sufficient detail (see specific comments below), and the results present only parts of the water balance on which the conclusions are then based. This results in a paper that is currently difficult to evaluate. Also, there is no discussion section present.

Reply: Implemented. The focus of the study is to measure the deep soil moisture infiltration in the Pinus sylvestris var. mongolica forest land, using a newly designed lysimeter. The detailed description of the design and application of the instrument has been documented in a previous publication on HESS (Cheng et al., 2017), so it is only briefly described in this study, see page 5-6, line 138-160. The Pinus sylvestris var. mongolica has been in existence in the study area for more than 30 years, so the purpose of this study is to find out whether there are sufficient water resource available in the region to support vegetation ecosystem, through the measurement of deep soil water recharge (or DSR). As suggested, the title of the paper has been changed to "On the soil moisture dynamics of sand-fixing Pinus sylvestris var. mongolica forest in a semi-arid region", which is a better representation of the body of this study. We have rewritten the discussion part as suggested. See page 13-14

Specific comments: Line 16: in this semi-arid region Line 18: "as an example" In the introduction I read in line 90-99 that many of the reforestation efforts are unsuccessful and Pinus sylvestris var. mongolica (Psvm) is the most common specie used in 3NSP. There is no statement on whether this species is more resilient, or why the abstract mention this as being an example. Can these several statements be more connected to clarify the actual success of using Psvm?

Reply: Implemented. Most of the afforestation practices mentioned in the introduction of the paper were terminated prematurely due to various reasons. In contrast, the Pinus sylvestris var. mongolica was found to be a suitable species in the most of practice of afforestation in the Three North Region of China and have been planted here for 30 years, but reforestation in some places with unclear reasons. Specifically, trees grow into dwarf trees, which are ineffective in battling land degradation, or even die, see

page 3, line 76-79. Therefore, the lessons leant from this afforestation practice are important for future adaptation of this species in other regions as well. We have revised the statements and provided further references to address the issue of resilience of this species. See page 2-3, line 64-79.

Line 29-30: Reported results are for period2016-2018, and it is concluded deep soil recharge happened and thus Psvm can sustainably develop. I think this conclusion is not merited on a 3-year observation period. There is nothing reported on the state of the 30-year-old Psvm forest, are these trees normally developed or not? Were they irrigated during that time? What is the minimum amount of water needed for these trees to survive? Also I have my doubts at the significance of the reported digits of the water balance. See later comments.

Reply: Implemented. The point raised here is similar to the comment No. 2 from the other reviewer. Please see our detailed response to that comment. "One should be cautious that the three years (2016-2018) soil moisture measurements presented in this study may not always be reliable for performing long term (such as decades long) prediction of whether the studied species can develop sustainably over decades as some artificial trees may have life cycles over decades long. Therefore, continuous (preferably decades long) measurements are necessary in the future. Another notable point is that the adaptability of long-lived woody species may not be based solely on water, temperature, light, and soil texture. Despite of such limitations, we think this three-year investigation offers an important step for understanding the soil moisture dynamics of sand-fixing Pinus sylvestris var. mongolica forest in a semi-arid region. Furthermore, these three years happen to encompass rather dramatically different weather patterns in the region (wet versus dry years), thus offer additional insights on the function of the Pinus sylvestris var. mongolica forest under highly variable external forces."

Line 37: In the abstract the term desertification was used in the context of an arid environment. Here it is used in a broader sense. The World Atlas of Desertification

(2018) has revised the definition due to confusion outside the context of (semi-)arid areas and now promotes the use of the term land degradation instead of desertification. Whether the authors use desertification or land degradation I suggest to refer to a formal definition in this particular general context. Line 101:"sustainable" This question is not veryspecific. Sustainable in term of what exactly?

Reply: Implemented. We have revised the language and the choice of words. For instance, we will use the term "land degradation", and we will re-define the term "sustainable" as meeting the growth needs of Pinus sylvestris var. mongolica and also having excess water to replenish deep soil layer as a sustainable standard. See page 3 line 87-89, page 13, line 306-308.

Line 128-129: Unclear if this refers to general observations from literature or from the inspected field site. Please explain. Line 135-137: "measurements were made without interference of canopy in the middle between forest lines" So, no root water uptake was measured, no intercepted precipitation or interecepted evaporation? The assumption being that the amount of deep recharge will be the amount available to the trees to take up water? But in that case you are missing the interception term, and the amount of water infiltration in the forest will be less than what is measurement in the DSR setup, which then invalidates your conclusions. Please explain more clearly so the reader can follow.

Reply: Implemented. "Root analysis of Pinus sylvestris var. mongolica shows that it has a shallow root distribution and underdeveloped main root, which belongs to a typical lateral root type" This is the result of in situ observation after on-site excavation. Similar findings can also be found in the literature. In semi-arid areas, vegetation depends on precipitation, and roots are concentrated in shallow soil. Measuring the water absorption of roots, precipitation or interception of the canopy are complicated processes and often involves a great degree of uncertainty. This research, however, regards the canopy and topsoil as an integrated entity. And by measuring the amount of water entering the entity (precipitation) and the amount of water leaving the entity

(deep soil recharge), one may conduct a water balance computation to calculate the amount of overall evapotranspiration.

Line 139-143: What was the soil composition/classification? Which soil moisture sensors were used? The miniTrase refers to the cable tester, not to the particular type of moisture sensor. Was a factory calibration used with the soil moisture sensors, and which one? Line 146: "sur-face runoff does not exist" Was this not observed, even in snowmelt conditions? What was the slope in the area? Please specify.

Reply: Implemented. The soil type in this area is sandy soil, the particle size distribution of 0-200 cm depth is as follows: extra coarse sand of 0.00%, coarse sand of 3.23%, middle sand of 50.53%, find sand of 36.06%, very fine sand of 7.19%, and silt sand of 2.99%. The EC-5 soil moisture probe was used and the correction equation is (wu, 2014ïijŻField-Specific Calibration and Evaluation of ECH 2 O EC-5 Sensor for Sandy Soils): . where xsand and ysand are Analog value and Standard value. The topographic variation of area is almost negligible and long-term observations show that there is no surface runoff.

Line 152-154: "point A and B" not indicated in Fig. 2 Fig 2: Does not include an arrow for evaporation? Is it not measured?

Reply: Implemented. The figure is revised as below

Fig 3: The caption states the figure shows annual precipitation as well, but only the soil moisture readings are reported. As soil moisture changes in the lysimeter are a result of precipitation and soil evaporation (assuming no trees were growing in the lysimeter), so please include precipiatation over 2016-2018 in the results section. Also in the frozen season the soil moisture sensors readings drop. As freezing impacts the dielectric permittivity the sensor readings can be impacted. See for example: Hallikainen, M. T., Ulaby, F. T., Dobson, M. C., El-Rayes, M. A., & Wu, L. K. (1985). Microwave dielectric behavior of wet soil-part 1: Empirical models and experimental observations. IEEE Transactions on Geoscience and Remote Sensing, (1), 25-34. Nothing is mentioned in

the text, regarding these readings.

Reply: Implemented. We have revised the manuscript to include more detailed information on the soil moisture data and precipitation data of the past three years (2016-2018) to analyze the DSR. Under low temperature conditions in the winter, the accuracy of EC-5 may drop by 5% (according to the original manufacturer's instructions). To avoid the possibly unreliable data in the winter, we focus on analyzing the data from April to November (unfrozen ground period).

Line 186: Please explain how I can assess this from Fig 3? I see more than four increases in soil moisture at the 200 cm depth. Do the authors mean there are four time when water was collected in the measurement section of the DSR?

Reply: Implemented. According to Figure 3, the soil moisture content of the upper 200 cm soil layer fluctuates multiple times during the three-year experimental period. After November, the soil moisture content of the upper 200 cm soil layer fluctuations but DSR is not detected. This is probably due to the error of the EC-5 probe under frozen winter condition. Therefore, the active research period has been revised to from April to November each year. Between April to November, the DSR signals generated by the precipitations on August 11 and August 22 cannot be distinguished from each other, so we combine them together as one event. We will revisit this matter in this revised version to get a better description by inspecting the data more carefully. Fig 4: Changes are reported in increments of <0.01%. What was the accuracy of the soil moisture sensors, and are the data in Fig 4 not impacted by this? Or is the scale perhaps not what I think I am seeing, fraction instead of percentage?

Reply: Implemented. Figure 4 is revised to avoid the confusion. The data on the abscissa will be multiplied by 100%, with a range of 0-0.12, or 0-12% for better inspection.

Line 201: "annual soil moisture infiltration" Please explain how these reported numbers were obtained in the methodology, I suspect the infiltration is either weighed or measured otherwise, but it is not described at present. Table 2: If the accuracy of the

precipitation measurement is 0.2 mm, the calculated ET cannot have more reported significant digits. Please adjust. Also if as stated the measurements were made in the middle between the forest line, what then caused the transpiration term in the ET? Or was the DSR placed underneath the canopy, while precipitation and soil moisture were measured outside of the canopy. For me as a reader it became a bit vague at this point. In this water balance the canopy interception is not considered as well, making the concluding statements doubtful.

Reply: Implemented. "The annual soil moisture infiltration is only 0.2 mm, which is 1.2 mm lower than that of 2016." Should be revised as "The annual DSR measured at the 200cm soil depth is only 0.2 mm, which is 1.2 mm lower than that of 2016."

Line 213: And this would not depend on posterior soil moisture conditions as well? Line 318&324: Please adjust the reported significant digits.

Reply: Implemented. The line 210-213 describes the general state of soil moisture fluctuations in the experimental area after precipitation. The soil moisture rises with precipitation and decreases after vegetation consumption or infiltration. It is notable that the precipitation event in arid and semi-arid areas is not always strong enough to induce DSR. The line 318-314 refers to the change in soil moisture in 2016 and this part will be revised accordingly.see line 255-260

Please also note the supplement to this comment:
https://www.hydrol-earth-syst-sci-discuss.net/hess-2019-110/hess-2019-110-AC4-supplement.pdf

**Supplement:**

**On the soil moisture dynamics of sand-fixing Pinus sylvestris var. mongolica forest 
[revised manuscript text omitted]
 land degradation in semi-arid regions of China? Or on the soil moisture dynamics of sand-fixing Pinus sylvestris var. mongolica forest in a semi-arid region. We will try to answer this question from a hydrological point of view by inspecting the relationship of precipitation, soil moisture change, and deep soil recharge (DSR) (referring to recharge that can reach a depth more than 200 cm and may eventually replenish the groundwater reservoir). In

85    particularly, if a sizable DSR can occur, meaning that groundwater may be recharged from precipitation, or there will be no dry soil layer below the root system layer in this region (Burgess et al., 1998;Bouma and Dekker, 1978). The sustainable reforestation in the region is possible, meaning that: 1) the precipitation mount meets the growth needs of Pinus sylvestris var. mongolica, and 2) the system still has excess amount of water to replenish deep soil layer as a sustainable standard. To accomplish this goal, this study uses a newly developed DSR lysimeter to monitor a 30-year

90    old pine artificial forest in the Northwest China. The collected dataset is used to understand the soil moisture dynamic and the DSR of the Pinus sylvestris var. mongolica in sandy land. Specifically, we try to tackle the following issues: 1) Sources of soil water recharge in semi-arid areas, especially the source of deep soil layer moisture; 2) The precipitation density that causes infiltration and its maximal penetrating depth; 3) The rate of annual precipitation infiltration; 4) The evaporation amount of the pine forestry land. The ultimate goal is to find out whether the rain-feed

95    Pinus sylvestris var. mongolica sand-fixing forest can develop sustainably or not in the study site.

**2 Material and Method**

**2.1 Overview of the study area**

The area under study is located in Chagan Naoer, on the northeastern edge of Mu Us Sandy land (39°05′16.2″N, 109°36′04″E), as shown in Figure 1. Mu Us Sandy land mainly consists of semi-fixed and fixed sand dunes, adjacent

100    with the Loess Plateau, located in a desert-loess transitional zone. It has northwestern wind in the winter with a typically dry winter climate and frequent sandstorm. It has southeastern monsoon in the summer. The summer climate is relatively humid, and it is easy to form local heavy precipitation. The multi-year average precipitation is 400 mm,

mostly concentrated during the summer. The groundwater table depth varies between 2 m to17 m in Mu Us Sandyland, and it is 8 m at the experimental area of this study (Runnström, 2003). The groundwater table is lower in the summer and higher in the spring, with a variation less than 1.5 m. Since the initiation of 3NSP at Mu Us Sandy land in 1989, Pinus sylvestris var. mongolica has been planted in lines in the experimental area, with a 10 m line spacing, an average plant height of 6 m, and an average crown diameter of 6.6 m. The seasonal frozen soil period in the experimental area is from January to April, and from November to December in an annual base (Li et al., 2013a). The soil type in this area is sandy soil, the particle size distribution of 0-200 cm depth is as follows: extra coarse sand of 0.00%, coarse sand of 3.23%, middle sand of 50.53%, find sand of 36.06%, very fine sand of 7.19%, and silt sand of 2.99%.

[Figure]

Figure 1. Geographic location of the experimental area and study site.

**2.2 Experimental design**

Root analysis of Pinus sylvestris var. mongolica shows that it has a shallow root distribution and underdeveloped main root, which belongs to a typical lateral root type. The 30-year-old Pinus sylvestris var. mongolica's root distribution can reach 6.5 m, with a concentrated area of 2.5 m. In this study site, the ground water level is too deep to supply for roots, so Pinus sylvestris var. mongolica root is mostly distributed over a vertical range of 0-0.5m and relies on precipitation for water supply. The original main root usually stops growing at 1.5-2 m depth (Zhu et al., 2006). Therefore, in the experimental design of this study, the lowest soil moisture sensor is placed at a depth of 200 cm, and the lysimeter placement is also at the 200 cm depth. The canopy of Pinus sylvestris var. mongolica is capable of intercepting precipitation, thus affects the measurements of precipitation and soil moisture directly underneath the canopy (Roth et al., 2007). Therefore, the measurements of this study are made in the middle between the forest lines, without the interference of canopy.

**2.2.1 Soil moisture monitoring**

Based on the root depth of Pinus sylvestris var. mongolica, the depth range of soil moisture sensor placement is determined. A soil section is cut out in the middle between two forest lines. The section consists of a layer of dead

tree leaves, a leached layer, a depositional layer, and a native soil layer, which is of fine sand. Soil moisture sensors (EC-5, accuracy 0.1% mm, USA) are placed in soil layers at 20 cm, 40 cm, 80 cm, 120 cm, 160 cm, and 200 cm depths. The measurement interval is one hour. The EC-5 soil moisture probes are used and the correction equation is (Wu et al., 2014):

$$y_{Sandy} = 1.0223x_{Sandy} - 0.0302 \quad (R^2 = 0.9098) \tag{1}$$

where $x_{sand}$ and $y_{sand}$ are analog value and standard value, respectively. The topographic variation of the area is almost negligible and long-term observations show that there is no surface runoff. Under low temperature conditions in the winter, the accuracy of soil moisture sensor (EC-5) may drop by 5% (according to the original manufacturer's instruction). To avoid the possibly unreliable data in the winter, this research focuses on analyzing the data from April to November (unfrozen ground period).

**2.2.2 DSR monitoring**

To study the moisture distribution of Pinus sylvestris var. mongolica in Mu Us Sandy land, two sets of data need to be collected: precipitation from a rain gauge, and DSR measurement from a lysimeter. This new lysimeter has a few innovations (see Figure 2) that can be outlined as follows. Instead of setting the upper boundary of the lysimeter at ground surface, the new design has its upper boundary at a designed depth (denoted as depth-A) where infiltration will be measured. A cylindrical container with a diameter of 20 cm to 40 cm with impermeable walls is installed from depth-A downward to a deeper depth-B. The length of AB is determined according to the capillary rise of the in-situ soil, which can be calculated using the average grain size of soils within AB. More specifically, the length of AB is greater than the capillary rise of soils within AB and it is usually great than 0.6 m (Liu et al., 2014). At the soil surface there is a device to measure the amount of precipitation and at the base of the instrument (depth B), a water collection device is used to measure the amount of water exiting the base downward. Surface runoff does not exist in the experimental area, thus is not a concern. Precipitation is monitored by rain gauge (Spectrum, USA, accuracy 0.2 mm), placed 1.5 m above ground surface. New lysimeter to measure the DSR (Cheng et al., 2017b). Such a new lysimeter has two parts: an upper water balance part and a lower measurement part. The advantage of this design is that when soil at point B in Figure 2 reaches saturation, the capillary water reaches point A. Therefore, when infiltrated water enters the water balance part at depth A, additional infiltrated water after satisfying the saturation of soil between A and B will go into the measuring part. The measurement part has a measurement accuracy of 0.2 mm (rain gauge, Spectrum, USA). The lysimeter is placed at 200 cm depth to measure DSR, meaning that any precipitation-induced infiltration passing the 200 cm depth will not be subject to evaportranspirative process anymore. In other words, the infiltrated water that can pass the 200 cm depth of soil will keep going down and may become groundwater recharge.

Before taking the measurements, the new lysimeter needs to be placed one year in advance, going through naturally settlement for a year, and allowing the soil to reach its pre-installation condition.

[Figure]

Figure 2. The schematic plot of a new lysimeter (on the right) with respect to the conventional lysimeter (on the left).

**3 Results**

The soil moisture variation of Pinus sylvestris var. mongolica in 2016 is shown in Figure 3. It reveals that soil moisture has obvious seasonal variational trends. The soil from January to March is frozen. The near surface soil moisture recharge is from snowmelt. When the near surface frozen soil starts to thaw, soil at the 20 cm depth is recharged on February 9th, 16th and 26th in 2016. Soil at depths greater than 20 cm remains relatively stable. Frequent precipitation events usually occur from June to November, during which soil moisture changes considerably, and soil moistures at different depths exhibit periodic increase or decrease, regulated by the interplay of precipitation and evapotranspiration. After February 26th in 2016, soil gradually thaws completely. Figure 3 shows that snowmelt can recharge the soil moisture as deep as 160 cm. The soil moisture at 200 cm depth is recharged for the first time after a heavy precipitation event on July 8th in 2016.

[Figure]

Figure 3. Annual precipitation and soil moisture of each layers in 2016.

According to Figure 3, the soil moisture content of the upper 200 cm soil layer fluctuates multiple times during the three-year experimental period. After November, the soil moisture content of the upper 200 cm soil layer fluctuations but DSR is not detected. This is probably due to the error of the EC-5 probe under frozen winter condition. Therefore, the active research period has been revised to from April to November each year.

In order to study the degree of soil moisture response to precipitation in individual layers, this research choose each layer's soil moisture at the beginning of each month of 2016 as a representative, to observe whether the soil moisture in a specific layer is recharged. Figure 4 shows the soil moistures at depths of 20 cm, 40 cm, 80 cm, 120 cm, 160 cm, and 200 cm at the beginning of each month. From Figure 3, the soil of Pinus sylvestris var. mongolica exhibits four distinctive layers: an evaporation layer at 0-40 cm depth, a lateral root activity layer at 40-160 cm depth, a dry soil layer at 160-200 cm depth, and a deep soil layer below 200 cm. For the 0-40 cm evaporation layer, the soil moisture increases only under the effect of precipitation or snowmelt. Its moisture content decreases rapidly under the interplay of evaporation and infiltration. For the 40-160 cm root activity layer, the soil moisture is recharged from infiltrated water passing through the upper layer, and it gradually decreases under the effects of infiltration and root moisture absorption. For the 160-200 cm dry soil, the infiltrated water hardly reaches this layer, and the soil layer with a soil moisture under the withered point is formed. The deep soil below 200 cm depth is of native sand soil, and Figure 3 shows that the soil moisture content of this layer is recharged five times under heavy precipitation events in 2016.

[Figure]

Figure 4. Month soil moisture changes of every soil layers, 2016.

[revised manuscript text omitted]

300     year (preferably a decade long) experiment.

**4 Discussion**

    In semi-arid areas, the main limiting factor for trees is available water resources (Gao et al., 2014b;Skarpe, 1991). Therefore, the key to understand the vegetation ecosystem in semi-arid areas is to study the supply of water resources (Cheng et al., 2018;Cheng et al., 2017a). The Pinus sylvestris var. mongolica has been in existence in the study area

305     for more than 30 years, so the purpose of this study is to find out whether there is sufficient water resource available in the region to support vegetation ecosystem, through the measurement of deep soil water recharge (or DSR). The "sustainable" growth of plants in this study means that water resource from precipitation can meet the growth needs of Pinus sylvestris var. mongolica, and can still have an excess amount of water to replenish deep soil layer (to recharge the deep groundwater system beyond the root zones of plants).  In this study, the soil moisture distribution of Pinus

310     sylvestris var. mongolica was studied by using the new designed lysimeter to measure whether the soil layer below the root layer could produce DSR or not.

    Lysimeter has been developed for hundreds of years and is widely used in the field of water balance research (Liu et al., 2002;López-Urrea et al., 2006;Boast and Robertson, 1982). The traditional Lysimeter has a size limitation, and

is ineffective in measuring trees and other large plants (Fritschen et al., 1977). This study used a newly designed Lysimeter to measure the deep soil moisture infiltration for water balance investigation (Cheng et al., 2017a).

Vegetation depends on precipitation, and roots are concentrated in shallow soil. Measuring the water absorption of roots, precipitation or interception of the canopy are complicated processes and often involves a great degree of uncertainty. This research, however, regards the canopy and topsoil as an integrated entity. And by measuring the amount of water entering the entity (precipitation) and the amount of water leaving the entity (deep soil recharge), one may conduct a water balance computation to calculate the amount of overall evapotranspiration.

This study monitored the distribution of soil moisture of Pinus sylvestris var. mongolica forest under the rain-fed conditions in the last three years, and verified whether soil moisture was sufficient (or sustainable) under existing precipitation conditions. One should be cautious that the three-year (2016-2018) soil moisture measurements presented in this study may not always be reliable for performing long term (such as decades long) prediction of whether the studied species can develop sustainably over decades as some artificial trees may have life cycles over decades long. Therefore, continuous (preferably decades long) measurements are necessary in the future. Another notable point is that the adaptability of long-lived woody species may not be based solely on water, temperature, light, and soil texture. Despite such limitations, this three-year investigation offers an important step for understanding the soil moisture dynamics of sand-fixing Pinus sylvestris var. mongolica forest in a semi-arid region. Furthermore, these three years (2016-2018)_happen to encompass rather dramatically different weather patterns in the region (wet versus dry years), thus offer additional insights on the function of the Pinus sylvestris var. mongolica forest under highly variable external forces.

**5 Conclusions**

Precipitation is almost the only water sources for replenishing the groundwater system in the Mu Us Sandy land, construction of a vast artificial shelter forest may have detrimental effect on ecological environment in arid regions and may substantially change the evapotranspiration pattern in the region and will greatly affect the groundwater recharge in the region, which directly determines if the water resource for reforestation. This study uses a 30-years-old mature Pinus sylvestris var. mongolica forest of a 10-meter line spacing as the target of the experiment. Using a new lysimeter to monitor DSR and to accurately calculate water balance from 2016 to 2018. The following conclusions can be drawn from this study:

1. Pinus sylvestris var. mongolica forest soil in high latitude region, as Mu Us Sandy land has two significant moisture recharge processes in an annual base: spring snow melted moisture infiltration-recharge process and summer precipitation-recharge process. The recharge depth of spring snow melted moisture recharge process can reach 160 cm of depth. The summer precipitation-recharge process results in DSR, recharging the soil moisture below 200 cm. The DSR of 2016-2018 is 1.4 mm, 0.2 mm, 1.2 mm, respectively. Under the existing precipitation conditions, water supply in the rain-fed pine forest can meet the consumption of vegetations but the remaining amount of rain-fed infiltration that can percolate into deep soil layer is small.

2. The experimental results show that the precipitation intensities are respectively 2.6 mm/d, 3.2 mm/d, 3.4 mm/d, 8.2 mm/d, 8.2 mm/d, and 13.2 mm/d when precipitation infiltrates into 20 cm, 40 cm, 80 cm, 120 cm, 160 cm, and 200 cm soil depths. Infiltration depth and precipitation intensity are not linearly related.

3. In semi-arid areas, the annual precipitation varies greatly, the dry and wet years alternate, DSR mount is relatively small, soil water mount is limited. The growth of Pinus is affected by annual precipitation. In 2016-2018, compared to the start of each year, soil moisture content increases 38.056 mm, -16 mm, 54.747 mm, and the negative value of year 2017 means that the soil moisture storage decreased by 16 mm.

4. The precipitations in 2016-2018 are 466.4 mm, 309 mm, 472.2 mm, and the associated DSR values are 1.4 mm, 0.2 mm, 1.2 mm, respectively. Under the current precipitation condition and reforestation design, the natural recharging moisture of Pinus sylvestris var. mongolica can meet the plant growth needs, and have additional moisture for DSR which may eventually recharge groundwater.

5. Calculation based on these dataset shows that the annual evaporation of Pinus sylvestris var. mongolica forest in Mu Us sandy land is 426.96 mm, 324.6 mm, 416.253 mm for year 2016-2018, respectively. Pinus automatically adjusts its evapotranspiration in response to different precipitation amount, and this may affect the development of Pinus sylvestris var. As extreme weather conditions happen more frequently worldwide (possibly due to the global warming effect), the arid region precipitation may change rapidly. Whether Pinus sylvestris var. mongolica can adapt to this trend is a question that still needs decades-long observational effort.

**Acknowledgements:**

This study was supported with research grants from the Fundamental Research Funds for the Central Universities (BLX201814) and the National Natural Science Foundation of China (41771306). I especially thanks Chinese Scholar Council support me to go to Texas A&M University as a visiting researcher. We sincerely thank two anonymous reviewers and Editor for their constructive and critical comments which help us improve the quality of the manuscript.